# Antibacterial and Antioxidant Activity of Synthetic Polyoxygenated Flavonoids

**DOI:** 10.3390/ijms25115999

**Published:** 2024-05-30

**Authors:** Mauricio Enrique Osorio-Olivares, Yesseny Vásquez-Martínez, Katy Díaz, Javiera Canelo, Lautaro Taborga, Luis Espinoza-Catalán

**Affiliations:** 1Laboratorio de Síntesis Orgánica, Departamento de Química, Universidad Técnica Federico Santa María, Valparaíso 2390123, Chile; luis.espinozac@usm.cl; 2Escuela de Medicina, Facultad de Ciencias Médicas, Laboratorio de Virología Molecular y Control de Patógenos, Facultad de Química y Biología, Universidad de Santiago de Chile, Santiago 9170022, Chile; yesseny.vasquez@usach.cl (Y.V.-M.); javiera.canelo@usach.cl (J.C.); 3Laboratorio de Pruebas Biológicas, Departamento de Química, Universidad Técnica Federico Santa María, Valparaíso 2390123, Chile; katy.diaz@usm.cl; 4Laboratorio de Productos Naturales, Departamento de Química, Universidad Técnica Federico Santa María, Valparaíso 2390123, Chile; lautaro.taborga@usm.cl

**Keywords:** flavones, flavanones, antibacterial, antioxidant

## Abstract

Flavonoids are an abundant class of naturally occurring compounds with broad biological activities, but their limited abundance in nature restricts their use in medicines and food additives. Here we present the synthesis and determination of the antibacterial and antioxidant activities of twenty-two structurally related flavonoids (five of which are new) by scientifically validated methods. Flavanones (**FV1**–**FV11**) had low inhibitory activity against the bacterial growth of MRSA 97-7. However, **FV2** (C5,7,3′,4′ = OH) and **FV6** (C5,7 = OH; C4′ = SCH**_3_**) had excellent bacterial growth inhibitory activity against Gram-negative *E. coli* (MIC = 25 µg/mL for both), while Chloramphenicol (MIC = 25 µg/mL) and **FV1** (C5,7,3′ = OCH_3_; 4′ = OH) showed inhibitory activity against Gram-positive *L. monocytogenes* (MIC = 25 µg/mL). From the flavone series (**FO1**–**FO11**), **FO2** (C5,7,3′,4′ = OH), **FO3** (C5,7,4′ = OH; 3′ = OCH_3_)**,** and **FO5** (C5,7,4′ = OH) showed good inhibitory activity against Gram-positive MRSA 97-7 (MIC = 50, 12, and 50 µg/mL, respectively), with **FO3** being more active than the positive control Vancomycin (MIC = 25 µg/mL). **FO10** (C5,7= OH; 4′ = OCH_3_) showed high inhibitory activity against *E. coli* and *L. monocytogenes* (MIC = 25 and 15 µg/mL, respectively). These data add significantly to our knowledge of the structural requirements to combat these human pathogens. The positions and number of hydroxyl groups were key to the antibacterial and antioxidant activities.

## 1. Introduction

Flavonoids are highly prevalent in plants as a product of their secondary metabolism and as a defense against biotic and abiotic stresses [1]. Flavanones are a subclass of flavonoids, with fruits and fruit juices of the *Citrus* genus being the main sources of flavanones, and they are responsible for important biological activities (Figure 1). These substances enhance the body’s defenses against oxidative stress and help the body prevent cardiovascular diseases, cancer, and atherosclerosis. Among them, a well-known antioxidant activity is associated with a reduced risk of chronic diseases in humans in diets rich in fruits and vegetables. In addition, they also show antiviral, anti-inflammatory, and antimicrobial activities [2]. Flavones are a subclass of flavonoids that are abundant in plants used as foods and beverages by humans (Figure 1). They are characterized by a double bond between C2 and C3 in the flavonoid backbone, no substitution at the C3 position, and oxidation at the C4 position. Flavones play several roles in plants, including as pigments, pesticides, UV protectants, and signaling molecules that promote root colonization by nitrogen-fixing bacteria and mycorrhizal fungi [3]. In humans, important biological activities have been reported; they improve the body’s defenses against oxidative stress and prevent cardiovascular diseases, atherosclerosis, and cancer. In addition, these compounds also display antiviral, anti-inflammatory, and antimicrobial properties [4,5,6].

Both flavanones and flavones can be easily synthesized from chalcones. Chalcones are obtained by Claisen–Schmidt condensation between benzaldehydes and acetophenone derivatives, mainly [7]. Flavanones are obtained from chalcones by cyclization in a mild alkaline medium [8] (Figure 1). Flavones are synthesized from the corresponding chalcones by oxidative cyclization in a non-aqueous alkaline medium [9]. Protection of the hydroxyl groups with chloromethyl methyl ether (MOMCl) is essential to avoid decompositions in alkaline media [10,11].

On the other hand, bacterial resistance is a public health problem growing dangerously worldwide and requires urgent measures in all governmental and social sectors [12]. *Escherichia coli* and *Staphylococcus aureus* are the two most concerning human pathogens associated with hospital-acquired and community-acquired infections, according to the most current global estimates of antibacterial resistance worldwide [12]. *Listeria monocytogenes* is a small Gram-positive, facultative intracellular rod bacteria anaerobe, non-spore-forming, invasive, and intracellular foodborne pathogen that is beta-hemolytic and catalase positive when grown on blood agar. It is known to cause a systemic disease called listeriosis in ruminants and humans [13], which is a relatively low-incidence disease, but it has a high case-fatality rate (20–30%) and causes serious diseases such as gastroenteritis, sepsis, encephalitis, and meningitis [14]. For this reason, it is necessary to look for new alternative molecules to commercial antibiotics or as co-adjuvants to these to reduce the possibility of generating resistance in these bacteria to antibiotics [15].

In light of the pharmaceutical industry’s lack of progress in the fight against antibiotic-resistant bacteria, a change of strategy is required. The industry’s focus on obtaining new and powerful antibiotics, which are increasingly expensive and have harmful side effects on the intestinal flora, has not ensured that bacteria will not create resistance over time. Flavonoids can alter several defense mechanisms in bacteria [16], causing the antibiotics to re-exert their inhibitory action or, in many cases, to restore their sensitivity to these antibiotics [17].

Most of the molecules for this study were selected based on previous research conducted by our team, which indicated their potential suitability for the purposes of this study [17].

In this work, we have evaluated the antibacterial activity of several known and five new (**FV6**, **FV7**, **FV9**, **FO6**, and **FV7**) flavonoids obtained synthetically by reactions known in the literature, with some modifications such as the introduction of SCH_3_ groups as a soft substituent at the 4′-position of the B Ring, longer reaction times, higher reaction temperatures, and the proportion of the reagents [9]. The synthesized flavonoids were evaluated against bacteria of major health concern in order to select the most active ones and thus propose them for studies of mechanisms of activity, to add some substituents to improve their bioavailability (acetylations, prenylations, etc.), and to have alternatives available as future drugs against resistant bacteria.

## 2. Results

### 2.1. Synthesis

The synthesis of chalcones (Figure 2), flavanones (Figure 3) and flavones (Figure 4) was carried out using known reaction mechanisms, previously protecting the hydroxyl groups in precursors with chloromethyl methyl ether (MOMCl), as shown in Figure 1 The yields of the chalcones are much higher than the subsequent reactions due to the presence of the OMOM protecting group, which withstands these conditions very well (Table 1). In general, flavones were obtained with lower yields than flavanones (Table 2), which may be attributed to the oxidative conditions required to achieve the closure of the C ring and form the corresponding double bond between C2 and C3.). The complete elucidation of the obtained structures was carried out by analyzing the 1D NMR spectra together with the 2D ones (HSQC and HMBC), not being necessary for other NOE-type experiments. All spectra and analyses are shown in the Appendix A.

### 2.2. In Vitro Antibacterial Effects of the Compounds on Human Pathogens

Among the flavonoids evaluated in the antibacterial activity assay (Table 3), only three flavanone compounds demonstrated significant inhibitory activity against bacterial growth: **FV1** against *L. monocytogenes* and **FV2** and **FV6** against *E. coli.* None of the compounds in this group were able to completely inhibit the growth of MRSA 97-7 at the concentration tested (50 µg/mL); however, three flavones (**FO2**, **FO3**, and **FO5**) were able to completely inhibit the growth; even **FO3** presented a MIC of 12 µg/mL lower than the positive control (Vancomycin^®^). All compounds that exhibited growth inhibition were specific for each bacterial pathogen.

The results show that the double bond at the C2-C3 position (flavones) and the hydroxyl groups at the C-5,7,4′ positions (R_2_,R_4_, R_7_ of **FO2**, **FO3**, and **FO5** in Figure 4) are important for antibacterial activity against MRSA 97-7 and *L. monocytogenes* (Gram-positive bacteria); however, the saturation of this bond (flavanones) is relevant for antibacterial activity against *E. coli* (Gram-negative bacteria), for example, compounds **FV2** and **FV6** (MIC = 25 µg/mL for both), whose MIC value is similar to the standard antibiotic Chloramphenicol (MIC = 25 µg/mL), although the activity of compound **FO10** against *E. coli* and *L. monocytogenes* stands out with MIC values of 25 µg/mL and 15 µg/mL, respectively. The presence of the two hydroxyl groups at 5 and 7 on the A-ring is indispensable for antibacterial activity, since the most active molecules possess these two free hydroxyl groups. The flavonoids are known to cause changes in the potency of the outer membrane and disruption in the inner membrane of the bacterial pathogen by an increased number of hydroxyl groups, which leads to an increase in the lipophilicity of the molecule, as in **FV2** and **FO2**, resulting in efficient uncouplers as they could transfer more protons per molecule. These same requirements are found in commercial fungicides to confer toxicity through the acidity of the hydroxyl group by uncoupling oxidative phosphorylation [18].

It is known that trihydroxylation at C5, C7, and C4′ is important for increasing the antibacterial activity of flavonoids [4]. However, in this work, we wanted to introduce the SCH_3_ group at C4′ to determine its influence on the activity, finding good activity against *E. coli* (MIC = 25 µg/mL) and regular activity against *L. monocytogenes* only in the case of flavanone (**FV6**). This SCH_3_ group in C4′ as a flavone (**FO6**) showed poor activity against the strains studied.

Our findings indicate that the flavone **FO10** (Acacetin), with hydroxyl substituents at C5 and C7 and an OCH_3_ group at C4′, is noted for its antibacterial activity against *E. coli* and *L. monocytogenes*. It has been demonstrated in the literature that Acacetin exhibits a broad spectrum of antibacterial activity, including against *S. aureus*, *E. faecalis*, *E. coli*, and *P. aeruginosa,* with MIC values for these bacteria ranging from 0.16 to 0.35 mg/mL [19], inhibits the virulence of *L. monocytogenes* [20], has a synergistic effect against MRSA strains in combination with Ampicillin and Oxicillin [21], as well as other interesting properties [22,23].

Flavones **FO2** (Luteolin) and **FO3** (Chrysoeriol) showed activity against *S. aureus* MRSA 97-7 (MIC = 50 and 12 µg/mL, respectively) and other Gram-positive as well as Gram-negative bacteria, which has been found in other reports [24,25]. It is known that changes in cell membrane permeability in *S. aureus* MRSA 97-7 after treatment with Luteolin increased the electrical conductivity according to the incubation time because Luteolin first enters the cell wall of the bacteria and then ruptures the cell membrane, causing an imbalance of ionic homeostasis that affected the metabolism of the pathogen, causing its death [26].

### 2.3. Antioxidant Activity of Synthesized Flavonoids

This work reiterates the significance of catechol groups in the development of potent DPPH free radical scavenging activity. According to Table 4, **FV2**, **FV9**, **FO2**, and **FO9** are the only active compounds. These compounds are distinguished by the presence of a catechol group on the B ring. This behavior is consistent with the findings of previous research in our laboratory [17,27]. The addition of SCH_3_ groups alone did not increase the antioxidant activity of **FV6**, **FV7**, **FO6**, and **FO7** molecules.

Compounds **FV2** and **FO2** are attributed, for their antibacterial and antioxidant properties demonstrated in this study, to the incorporation of four free hydroxyl groups into their structure (Table 2, Entry 2), two hydroxy groups in the A ring (C-5, 7), and one 1,2-dihydroxybenzene group in the B ring (C-3′, 4′), giving them conditions that improve their interactions in active sites. It has been found in previous studies that flavonoids act differently from commercial antibiotics [28]. It has been demonstrated that antibiotics generate reactive oxygen species (ROS) when they penetrate bacteria, while flavonoids do not produce oxidative stress in these pathogens. This suggests that they act as antibacterials by other mechanisms, for example, through the inhibition of pumps of expulsion such as ABC transporters. This enables them to enhance the action of the commercial antibiotic, as proposed by Wagner et al. (2009), who highlighted that the combined effect of phytodrugs with antimicrobial agents can result in alterations to the permeability of bacterial cell walls, the inhibition of efflux pumps, and the suppression of bacterial enzymes. These mechanisms collectively facilitate the delivery of antimicrobial agents to their intended targets [29]. Flavonoids with antioxidant activity may be able to counteract the oxidative stress of commercial antibiotics by decreasing their cytotoxicity. Furthermore, they could be used as food additives to protect against the oxidative degradation of foods by free radicals [30].

## 3. Materials and Methods

### 3.1. Chemistry

#### 3.1.1. General Data

All chemical reagents obtained were purchased from Merck (Darmstadt, Germany), Sigma-Aldrich (St. Louis, MO, USA), or Alfa Aesar (Kandel, Germany), were of the highest commercially available purity, and were used without previous purification. Melting points (mp: °C) were measured on a melting point apparatus (Stuart-Scientific SMP3) and are uncorrected. IR spectra were recorded as a KBr disk in a Thermo Scientific Nicolet 6700 FT-IR spectrometer (San Jose, CA, USA), and frequencies are reported in cm^−1^. High-resolution mass spectra were recorded on a compact QTOF MS + Elute UHPLC (HRMS-ESI) Bruker Daltonics (Bruker, Bremen, Germany). The analysis for the reaction products was performed with the following relevant parameters: dry temperature, 180 °C; nebulizer, 0.4 bar; dry gas, 4 L/min; and spray voltage, 4.5 kV in positive mode. Accurate mass measurements were performed at a resolving power of 140,000 FWHM in the range *m*/*z* 50–1300. ^1^H-, ^13^C-(DEPT 135), 2D HSQC, and 2D HMBC spectra were recorded in DMSO-*d*_6_ and CDCl_3_ solutions and referenced to the residual peaks of DMSO at δ 2.50 ppm and CDCl_3_ at δ 7.26 ppm for ^1^H and δ 39.5 ppm and 77.2 ppm for ^13^C, respectively, on a Bruker Avance Neo 400 Digital NMR spectrometer (Bruker, Rheinstetten, Germany), operating at 400.1 MHz for ^1^H and 100.6 MHz for ^13^C. Chemical shifts are reported in δ ppm, and coupling constants (*J*) are given in Hz. Silica gel (Merck 200–400 mesh, Merck, Darmstadt, Germany) was used for flash chromatography and silica gel plates HF-254 for thin layer chromatography (TLC). TLC spots were detected both under a UV lamp and after heating in 10% H_2_SO_4_ in H_2_O. Antioxidant determinations were performed on a Thermo Scientific Multiskan GO 96-well plate photometer (Vantaa, Finland).

#### 3.1.2. General Experimental Procedure for the Synthesis of Flavanones and Flavones

##### *O*-Protection with MOMCl of Acetophenone and Benzaldehyde Derivatives

*O*-protection was a crucial aspect in the synthesis of polyoxygenated chalcones in an alkaline medium, as oxidative reactions were observed under these conditions. A solution of the phenol derivative (1.0 eq) in CH_2_Cl_2_ (0.1–0.5 M concentration) was prepared and then added dropwise at 0 °C to a solution of i-Pr_2_NEt (1 eq × hydroxyl group) and MOMCl (1.5 eq × hydroxyl group). The reaction mixture was then stirred at room temperature until the reaction was complete. Following the slow addition of a saturated NH_4_Cl solution at 0 °C, the mixture was extracted with CH_2_Cl_2_ (15 mL × 3). The organic layers were combined, washed with brine, dried over Na_2_SO_4_, filtered, and concentrated under reduced pressure. The crude product was purified by flash column chromatography to afford MOM ether [10,11].

##### General Procedure for Preparation of Chalcones **CH1**–**CH11**

A solution of *O*-MOM-protected acetophenone (1 eq) and KOH (2.0 g dissolved in 10 mL of methanol, 6 eq) was prepared, to which *O*-MOM-protected benzaldehyde (1.1 eq) was added. The mixture was stirred at room temperature for a period of 24 h. The resulting mixture was then quenched in ice-cold water and acidified with 1 N HCl. The crude product was extracted with ethyl acetate (3 × 30 mL), and the combined extracts were washed with water (2 × 50 mL). The organic layer obtained after extraction was dried over anhydrous Na₂SO₄, filtered, and the solvent evaporated under reduced pressure. Purification of the crude mixture was achieved by silica gel column chromatography using a hexane-ethyl acetate solvent system, which afforded the chalcone [11].

The synthetic route to obtain chalcones is shown in Figure 1.

##### General Procedure for Preparation of Flavones **FO1**–**FO11**

A solution of chalcone (1 eq) and iodine (1 eq) was prepared in pyridine (1 mL per mmol of chalcone). The mixture was heated to 120 °C and stirred for 24 h. Once the reaction was complete (TLC analysis), the solution was carefully poured onto crushed ice and washed with 10% Na_2_S_2_O_3_ (30 mL) to remove iodine. The product was extracted with ethyl acetate (20 mL × 3) and washed with 0.1 N hydrochloric acid and water. The organic extract was washed with brine, dried over anhydrous sodium sulfate, and filtered. The solvent was removed under reduced pressure. The residue was purified by silica gel column chromatography using hexane-ethyl acetate to afford flavone [9].

##### General Procedure for Preparation of Flavanones **FV1**–**FV11**

A stirred solution of chalcone (1 eq) in ethanol (5 mL/mmol) was prepared, to which sodium acetate (7 eq) and water (equivalent to the volume of ethanol) were added. The reaction mixture was heated to reflux for 16 h and then allowed to cool to room temperature. The mixture was diluted with water and extracted with acetic acid. The combined organic phases were washed with brine, dried over anhydrous sodium sulfate, and filtered. The solvent was evaporated under reduced pressure. The residue was purified by column chromatography using hexane-ethyl acetate as a solvent system to afford flavanone [8,11]. Finally, we proceed to hydrolyze -OMOM groups with aqueous 1N HCl/ethanol/isopropanol (1:1:1) at reflux for 30 min to produce flavones **FO1**–**FO11** and flavanones **FV1**–**FV11**. The products were extracted with ethyl acetate (20 mL × 3) and subsequently washed with water. The organic extracts were then washed with brine and dried over anhydrous sodium sulfate. The solvent was then removed under reduced pressure. The mixture was then subjected to silica gel flash column chromatography (ethyl acetate/hexane mixtures were used as mobile phases) to obtain pure products.

Reactions for the preparation of synthetic flavones and flavanones are shown schematically in Figure 1.

All structures were confirmed by IR and NMR spectra, as discussed below.

#### 3.1.3. Physical Data of Synthesized Compound

Only the data on the new compounds is shown here. Data for all synthesized compounds are shown in the Appendix A along with images of 1D and 2D NMR, IR, and MS spectra.

##### Flavanones

5,7-dihydroxy-2-(4-(methylthio)phenyl)chroman-4-one (**FV6**) was obtained as racemic mixtures because these were synthesized from chalcone (**C6**) (5.1 mmol), NaOAc (35.7 mmol), and EtOH/H_2_O, then heated to reflux as described above. This crude was then hydrolyzed with a mixture of aqueous HCl (1 N)/ethanol/isopropanol at reflux and then purified by column chromatography using ethyl acetate in hexane in a gradient system (0–50% ethyl acetate in hexane) as the mobile phase to afford **FV6** as a white powder (591.7 mg, 51% from chalcone **C6**); mp: 236.8–238.0 °C; HRMS *m*/*z*, observed: 303.0691; C_16_H_14_O_4_S [M + H]^+^ requires: 303.0686. IR (film): ν_max_ cm^−1^: 3166, 2919, 2840, 1635, 1600, 1493, 1434, 1341, 1311, 1299, 1176, 1161. ^1^H-NMR (DMSO-*d*_6_) δ ppm: 12.11 (s, 1H, ArO*H*-5); 10.82 (s, 1H, ArO*H*-7); 7.44 (d, 2H, *J* = 8.2 Hz, Ar*H*-3′,5′); 7.30 (d, 2H, *J* = 8.2 Hz, Ar*H*-2′,6′); 5.91 (d, 1H, *J* = 1.6 Hz, Ar*H*-8); 5,89 (s, 1H, Ar*H*-6); 5.54 (dd, 1H, *J* = 12.4 and 2.6 Hz, C*H*-2); 3.24 (dd, 1H, *J* = 17.1 and 12.5 Hz, C*H*H-3); 2.75 (dd, 1H, *J* = 17.1 and 3.0 Hz, CH*H*-3); 2.48 (s, 3H, C*H*_3_S-4′). ^13^C-NMR (DMSO-*d*_6_) δ ppm: 196.0 (4-*C*=O); 166.7 (Ar*C*-7); 163.5 (Ar*C*-5); 162.7 (Ar*C*-1a); 138.7 (Ar*C*-4′); 135.1 (Ar*C*-1′); 127.3 (Ar*C-*3′,5′); 125.8 (Ar*C*-2′,6′); 101.8 (Ar*C*-4a); 95.9 (Ar*C*-6); 95.0 (Ar*C*-8); 78.0 (*C*H-2); 41.9 (*C*H_2_-3); 14.6 (C*H*_3_S-4′).

5-hydroxy-2-(4-(methylthio)phenyl)chroman-4-one (**FV7**) was obtained as racemic mixtures because these were synthesized from chalcone (**C7**) (1.82 mmol), NaOAc (12.7 mmol), and EtOH/H_2_O, then under-heated to reflux as described above. This crude was then hydrolyzed with a mixture of aqueous HCl (1 N)/ethanol/isopropanol at reflux and then purified by column chromatography using ethyl acetate in hexane in an isocratic system (50% ethyl acetate in hexane) as the mobile phase to afford **FV7** as a white powder (200.1 mg, 58% from chalcone **C7**); mp: 114.9–117.0 °C; HRMS *m*/*z*, observed: 287.0738; C_16_H_14_O_3_S [M + H]^+^ requires: 287.0736. IR (film): ν_max_ cm^−1^: 3421, 2923, 1667, 1621, 1576, 1465, 1352, 1338, 1205, 1049, 825, 729. ^1^H-NMR (DMSO-*d*_6_) δ ppm: 11.74 (s, 1H, ArO*H*-5); 7.45-7.49 (m, 3H, Ar*H*-3′,5′,7); 7.32 (d, 2H, *J* = 8.3 Hz, Ar*H*-2′,6′); 6.53 (dd, *J* = 7.8 and 6.8 Hz, Ar*H*-6,8); 5.65 (d, 1H, *J* = 12.8 and 2.8 Hz, C*H*-2); 3.41 (dd, 1H, *J* = 17.2 and 12.9 Hz, C*H*H-3); 2.88 (dd, 1H, *J* = 17.2 and 3.0 Hz, CH*H*-3); 2.49 (s, 3H, C*H*_3_S-4′). ^13^C-NMR (DMSO-*d*_6_) δ ppm: 198.7 (*C*-4); 161.2 (Ar*C*-5); 161.2 (Ar*C*-1a); 138.9 (Ar*C-*4′); 138.4 (Ar*C*-7); 134.9 (Ar*C*-1′); 127.3 (Ar*C*-3′,5′); 125.8 (Ar*C*-2′,6′); 108.8 (Ar*C*-6); 107.8 (Ar*C*-4a); 107.5 (Ar*C*-8); 78.1 (*C*H-2); 42.5(*C*H_2_-3); 14.6 (*C*H_3_S-4′).

2-(3,4-dihydroxyphenyl)-5,7-dimethoxychroman-4-one (**FV9**) was obtained as racemic mixtures because these were synthesized from chalcone (**C9**) (0.93 mmol), NaOAc (6.49 mmol), and EtOH/H_2_O, then heated to reflux as described above. This crude was then hydrolyzed with a mixture of aqueous HCl (1 N)/ethanol/isopropanol at reflux and then purified by column chromatography using ethyl acetate in hexane in an isocratic system (70% ethyl acetate in hexane) as the mobile phase to afford **FV9** as a pale yellow powder (79.2 mg, 27% from chalcone **C9**); mp: 172.5–173.3 °C (dec.); HRMS *m*/*z*, observed: 317.1025; C_17_H_16_O_6_ [M + H]^+^ requires: 317.1020. IR (KBr): ν _max_ cm^−1^: 3396, 2977, 2932, 2840, 1645, 1609, 1571, 1518, 1455, 1426, 1388, 1346, 1280, 1218, 1200, 1161, 1118, 1073, 811, 788, 556. ^1^H-NMR (DMSO-*d*_6_) δ ppm: 9.03 (s, 2H, Ar-O*H*-3′,4′); 6.85 (s, 1H, Ar*H*-6′); 6.72 (s, 2H, Ar*H*-2′,5′); 6.17 (s, 2H, Ar*H*-6,8); 5.31 (dd, 1H, *J* = 12.3 and 2.3 Hz, C*H*-2); 3.78 (s, 3H, C*H*_3_O-5); 3.75 (s, 3H, C*H*_3_O-7); 2.96 (dd, 1H, *J* = 16.2 and 12.6 Hz, CH*H*-3); 2.53 (dd, 1H, *J* = 16.4 and 2.6 Hz, C*H*H-3). ^13^C-NMR (DMSO-*d*_6_) δ ppm: 188.1 (4-*C*=O); 165.3 (Ar*C*-7); 164.4 (Ar*C*-5); 161.7 (Ar*C*-8a); 145.6 (Ar*C*-4′); 145.2 (Ar*C*-3′); 129.7 (Ar*C-*1′); 117.8 (Ar*C*-6′); 115.3 (Ar*C*-5′); 114.2 (Ar*C*-2′); 105.4 (Ar*C*-4a); 93.7 (Ar*C*-6); 92.7 (Ar*C*-8); 78.3 (*C*H-2); 55.8 (*C*H_3_O-5); 55.7 (*C*H_3_O-7); 44.8 (*C*H_2_-3).

##### Flavones

5,7-dihydroxy-2-(4-(methylthio)phenyl)-4*H*-chromen-4-one (**FO6**) was obtained from chalcone **C6** (2.89 mmol), using iodine (2.89 mmol), potassium iodide (2.89 mmol) and pyridine as solvent (8 mL), then heated at 120 °C for 14 h, as described above. This crude was then hydrolyzed with a mixture of aqueous HCl (1 N)/ethanol/isopropanol at reflux and then purified by column chromatography using ethyl acetate in hexane in a gradient system (0–40% ethyl acetate in hexane) as the mobile phase to afford **FO6** as a yellow solid (121.5 mg, 14% from chalcone **C6**); mp: 230–232 °C; HRMS *m*/*z*, observed: 301.0527; C_16_H_12_O_4_S [M + H]^+^ requires: 301.0529; IR (KBr): ν _max_ cm^−1^: 2921, 2849, 2705, 2622, 1655, 1610, 1563, 1501, 1487, 1424, 1354, 1278, 1249, 1168, 1101, 1029, 1012, 908, 851, 826, 806, 748, 729, 678, 525. ^1^H-NMR (DMSO-*d*_6_) δ ppm: 12.86 (s, 1H, ArO*H*-5); 10.8 (s, 1H, ArO*H*-7); 7.98 (d, 2H, *J* = 8.6 Hz, Ar*H*-2′,6′); 7.40 (d, 2H, *J* = 8.6 Hz, Ar*H*-3′,5′); 6.93 (s, 1H, C*H*-3); 6.50 (d, 1H, *J* = 2.0 Hz, Ar*H*-8); 6.20 (d, 1H, *J* = 2.0 Hz, Ar*H*-6); 2.54 (s, 3H, C*H*_3_S-4′). ^13^C-NMR (DMSO-*d*_6_) δ ppm: 181.7 (4-*C*=O); 164.3 (Ar*C*-7); 162.9 (Ar*C*-2); 161.4 (Ar*C*-5); 157.3 (Ar*C*-8a); 144.0 (Ar*C*-4′); 126.7 (Ar*C-*2′,6′); 126.6 (Ar*C*-1′); 125.5 (Ar*C*-3′,5′); 104.3 (*C*H-3); 103.9 (Ar*C*-4a); 98.9 (Ar*C*-6); 94.0 (Ar*C*-8); 14.0 (*C*H_3_S-4′).

5-hydroxy-2-(4-(methylthio)phenyl)-4H-chromen-4-one was (**FO7**) obtained from chalcone **C7** (1.71 mmol) using iodine (1.71 mmol), potassium iodide (1.71 mmol) and pyridine as solvent (6 mL), then under-heated at 120 °C for 10 h, as described above. This crude was then hydrolyzed with a mixture of aqueous HCl (1 N)/ethanol/isopropanol at reflux and then purified by column chromatography using methylene chloride in an isocratic system (100% methylene chloride) as the mobile phase to afford **FO7** as a yellow solid (286.9 mg, 59% from chalcone **C7**); mp: 171.2–172.2 °C (lit. [31] 166–167 °C); HRMS *m*/*z*, observed: 287.0738; C_16_H_14_O_3_S [M + H]^+^ requires: 287.0736; IR (KBr): ν _max_ cm^−1^: 3447, 3066, 2920, 1651, 1615, 1597, 1582, 1471, 1418, 1361, 1301, 1261, 1229, 1095, 1057, 996, 820, 800, 752. ^1^H-NMR (DMSO-*d*_6_) δ ppm: 12.70 (s, 1H, ArO*H*-5); 8.03 (d, 2H, *J* = 8.6 Hz, Ar*H*-2′,6′); 7.67 (dd, 1H, *J*_1_ = 8.4 and 8.3 Hz, Ar*H*-7); 7.42 (d, 2H, *J* = 8.6 Hz, Ar*H*-3′,5′); 7.19 (dd, 1H, *J*_1_ = 8.9 and 0.6 Hz, Ar*H*-8); 7.08 (s, 1H, C*H*-3); 6.80 (d, 1H, *J* = 7.7 Hz, Ar*H*-6); 2.55 (s, 3H, C*H*_3_S-4′). ^13^C-NMR (DMSO-*d*_6_) δ ppm: 183.1 (4-*C*=O); 163.9 (Ar*C*-2); 159.8 (Ar*C*-5); 155.8 (Ar*C*-8a); 144.6 (Ar*C*-4′); 135.9 (Ar*C*-7); 126.9 (Ar*C-*2′,6′); 126.4 (Ar*C*-1′); 125.5 (Ar*C*-3′,5′); 111.0 (Ar*C*-6); 110.1 (Ar*C*-4a); 107.5 (Ar*C*-8); 104.8 (*C*H-3); 14.0 (*C*H_3_S-4′).

### 3.2. Biological Assays

#### 3.2.1. In Vitro Antibacterial Activity Assays: Human Pathogens

##### Minimum Inhibitory Concentration Assay

A clinical isolate of methicillin-resistant *S. aureus* (97-7) was kindly donated by Dr. Marcela Wilkens from the Universidad de Santiago de Chile. *L. monocytogenes* 35153 and *Escherichia coli* ATCC 25922 are clinical isolates that belong to the Biological Tests Laboratory collection (Chemistry Department, Universidad Técnica Federico Santa María).

The antibacterial activities of the synthesized compounds (**FV1**–**FV11** and **FO1**–**FO11**) against methicillin-resistant *S. aureus* (MRSA) 97-7, *E. coli* ATCC 25922, and *L. monocytogenes* ATCC 35153 strains were tested using a modified serial dilution method that tested all the compounds in a concentration range between 0 and 50 μg/mL [32]. Briefly, stock solutions of compounds in DMSO were two-fold diluted in Mueller–Hinton Broth (MHB, Difco, Detroit, MI, USA). The final concentration of DMSO was ≤1.0% and did not affect the bacterial growth. The obtained solution was added to MHB and serially two-fold diluted in a 96-well microplate. An equal volume (1.5 μL) of bacterial suspension containing 10^6^ CFU/mL was inoculated into sterile 96-well microplates and incubated aerobically at 37 °C for 24 h on a shaker at 120 rpm. Percent inhibition of bacterial growth (PIAB) was calculated according to OD600 readings obtained from a Thermo Scientific Multiskan GO 96-well plate spectrophotometer (Waltham, MA, USA) [33]. The MIC was defined as the lowest concentration of compounds resulting in the complete inhibition of visible growth in petri dishes [34].

The positive control consisted of chloramphenicol (CALBIOCHEM; San Diego, CA, USA), ciprofloxacin (AK Scientific; Union City, CA, USA), and vancomycin (ChemCruz; Dallas, TX, USA) which were used at the same concentration gradient. The negative control was a 1% DMSO solution with an inoculum condition. Furthermore, an additional control was employed, consisting of 1% dimethyl sulfoxide (DMSO) without bacteria, in order to subtract background OD600 values. Each concentration of the compounds was tested in triplicate, and the values presented represent the mean ± SD of two independent experiments.

### 3.3. General Procedure to Determine the DPPH Radical Scavenging Activity

The radical scavenging activity of the prenylated compounds and starting materials towards the 2,2-diphenyl-1-picrylhydrazyl (DPPH) radical was quantified in accordance with the previously described methodology [35]. The assay was adapted to a screenable format on 96-well plates. In summary, stock solutions of each compound were prepared in methanol at a concentration of 1 mM (10 mL). The stock solutions were diluted to produce a range of concentrations between 1 and 200 µM. A volume of 90 µL of methanol was added to the 96-well plate, followed by 150 µL of the appropriate dilution. Finally, 60 µL of DPPH (Sigma-Aldrich, 0.5 mM) in methanol was added, resulting in a final concentration of 0.1 mM of DPPH. Methanol was employed as the blank sample. The mixtures were then incubated at room temperature for 30 min, after which their absorbencies were measured at 517 nm. Trolox was employed as the standard antioxidant. The radical-scavenging activity was calculated according to the following formula: The percentage inhibition was calculated as follows:

% Inhibition = [(blank absorbance − sample absorbance)/blank absorbance] × 100. The mean of the three IC50 values (concentration causing 50% inhibition) for each compound was determined graphically.

## 4. Conclusions

Our results demonstrated that synthesized flavanones have low inhibitory activity for the bacterial growth of MRSA 97-7; however, **FV2** and **FV6** have outstanding bacterial growth inhibitory activity against *E. coli* (MIC = 25 µg/mL for both), like Chloramphenicol (MIC = 25 µg/mL), and **FV1** possesses inhibitory activity against *L. monocytogenes* (MIC = 25 µg/mL). Synthesized flavones **FO2**, **FO3**, and **FO5** showed good inhibitory activity against MRSA 97-7 (MIC = 50, 12, and 50 µg/mL, respectively), with **FO3** being more active than the positive control Vancomicin (MIC = 25 µg/mL). **FO10** exhibited high inhibitory activity against *E. coli* and *L. monocytogenes* (MIC = 25 and 15 µg/mL, respectively). These data contribute significantly to our knowledge of the structural requirements to combat these human pathogens.

Flavonoids with *ortho*-dihydroxybenzene (catechol) groups are active as potent antioxidants. Flavones are more potent antioxidants than the corresponding flavanones.

New biological experiments are needed to determine the mechanisms of action of the most active substances as antibacterials.

The incorporation of the SCH_3_ group in flavanones (**FV6**) improves the inhibitory activity against Gram-negative bacteria (*E. coli*) with a value of MIC = 25 µg/mL compared to the methoxyl group (**FV10**) at the same position (MIC > 50 µg/mL). In the case of the respective flavones (**FO6** and **FO10**), the inverse relationship is obtained for *E. coli*.The results obtained in these studies justify the incorporation of other substituents to the flavonoids that proved to be more active as antibiotics, for example, C-prenylations.

## Data Availability

Data is contained within the article.

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
