# Peer review of "Antibacterial and Antioxidant Activity of Synthetic Polyoxygenated Flavonoids"

_ijms, 2024, doi:10.3390/ijms25115999_

Round 1
Reviewer 1 Report
Comments and Suggestions for Authors
The article concerns the synthesis of 18 known and 4 new flavonoids with a flavone and flavanone skeleton and the assessment of the ability of these compounds to scavenge free radicals and have antimicrobial activity. The synthesis involved the commonly used Cleisen-Schmidt condensation of appropriate acetophenones and benzaldehydes with protected hydroxyl groups to chalcones, followed by their oxidative cyclization. The new compounds, two flavones and two flavanones, had a thiomethyl group at the C4' position.
Studies on the ability of the obtained flavonoids to scavenge free radicals (DPPH) confirmed the well-known relationship that the C3'-C4' catechol moiety significantly increases the activity of flavones and flavanones in this respect.
Studies on the antimicrobial activity against Escherichia coli, Listeria monocytogenes and methicillin-resistant Staphylococcus aureus bacteria showed high activity of five flavonoids, including one new flavanone, which is a sulfur analogue of isosacuranetin. I consider the most interesting element of the article to be the discovery that replacing the hydroxyl group with a methylthiol group in the C4' position of flavanones (FV6 and FV10) significantly increases the antimicrobial activity against E. coli, while in the analogous pair of flavones (FO6 and FO10) there is no such relationship and acacetin is the stronger antibacterial factor.
The article would certainly become more attractive if, instead of the antioxidant activity, which has been previously determined many times for known compounds, and is very predictable for new ones, the Authors found another potential application of their new four compounds.
Other comments:
The summary should be informative as an independent text, therefore it should not use symbols for compounds that do not indicate their chemical structure. Please mention that 4 of the 22 compounds are new.
line 30, "different hydroxyl groups" - rather number of ..., position of...
line 36, „Flavones are a subclass……(Fig. 1)” - the sentence suggests that this group of flavonoids occurs only in citrus fruits, but this is not true
Fig. 1, the drawing is incorrect due to the convention for drawing chemical compounds, letters are not used in place of chemical bonds. It is also unnecessary because Fig. 4 and 5 show the skeletons of flavanones and flavones
Fig. 2, transfer to the supplement or remove
Line 74, remove "(Kim et al., 2022)"
Scheme 1, chalcones should be given abbreviations other than those used (C1, C2,...) because their use may be misleading. Authors use the same notations to indicate the position in the flavonoid skeleton.
Scheme 1, Figure 3, 4, Table 1,2, 3 – simplfy, if in all the compounds discussed in a specific position there is the same substituent (here, the hydrogen atom in positions C3', C5' in chalcones and C6 and C8 in flavanones and flavones, respectively), then it should be visible in the figures and scheme and removed from the table
Line 224 Daltonics not „Daltonik”
Line 226 bar not „Bar”
Line 233 expand the abbreviation "C.C."
Line 242 use "equivalent" or "eq" throughout the text, not "equiv"
Author Response
Response to Reviewer 1 Comments
- Summary
Thank you very much for taking the time to review this manuscript. Please find the detailed responses below and the corresponding corrections highlighted and in track changes in the re-submitted files.
All suggestions and comments were duly considered as follows:
R1: The summary should be informative as an independent text, therefore it should not use symbols for compounds that do not indicate their chemical structure. Please mention that 4 of the 22 compounds are new.
Response: The abstract was modified under suggestions and highlighted.
line 30, "different hydroxyl groups" - rather number of ..., position of...
Response: see new line 32.
line 36, „Flavones are a subclass……(Fig. 1)” - the sentence suggests that this group of flavonoids occurs only in citrus fruits, but this is not true.
Response: see new line 39-40.
Fig. 1, the drawing is incorrect due to the convention for drawing chemical compounds, letters are not used in place of chemical bonds. It is also unnecessary because Fig. 4 and 5 show the skeletons of flavanones and flavones.
Response: The drawing in fig.1 was modified as suggested.
R1: Fig. 2, transfer to the supplement or remove
Response: Fig. 2 was removed to supplementary information.
R1: Line 74, remove "(Kim et al., 2022)"
Response: Removed.
R1: Scheme 1, chalcones should be given abbreviations other than those used (C1, C2,...) because their use may be misleading. Authors use the same notations to indicate the position in the flavonoid skeleton.
Response: A new scheme 1 was included under suggestions.
R1: Scheme 1, Figure 3, 4, Table 1,2, 3 – simplfy, if in all the compounds discussed in a specific position there is the same substituent (here, the hydrogen atom in positions C3', C5' in chalcones and C6 and C8 in flavanones and flavones, respectively), then it should be visible in the figures and scheme and removed from the table.
Response: Tables and figures were simplified as suggested. See new file.
R1: Line 224 Daltonics not „Daltonik”
Response: Modified line 217
R1: Line 226 bar not „Bar”
Response: Modified line 219
R1: Line 233 expand the abbreviation "C.C."
Response: Modified in line 226 as “flash chromatography”
R1: Line 242 use "equivalent" or "eq" throughout the text, not "equiv"
Response: Modified in all text
Best regards
Dr. Mauricio Osorio
Reviewer 2 Report
Comments and Suggestions for Authors
The article "Antibacterial and antioxidant activity of synthetic polyoxygenated flavonoids" describes the synthesis of 11 flavones and 11 flavanones (4 of them are new) and their biological activities as antimicrobials against a selection of gram-positive and gram-negative bacteria. Additionally, antioxidant activities are also analyzed using the DPPH radical method.
In my opinion, the synthetic part used in this article is widely known, and the novelty provided at this point is somewhat limited. On the other hand, the structural characterization of all the synthesized final compounds and intermediate chalcones is adequately described in the supplementary material, with high purity of all compounds confirmed by NMR and HRMS for the new compounds.
The antimicrobial activity of the synthesized compounds is where the main novelty of this article lies. The percentage of inhibition at 50 μg/mL along with their MIC against three bacteria, MRSA 97-7, E. coli, and L. monocytogenes, is described, with positive results found for compounds FV1, FV2, FV6, FO3, and FO10. Despite being promising results, in my opinion, it would be necessary to delve a little deeper into these antimicrobial activity results by conducting some antibiofilm studies, both inhibition and disruption of the preformed biofilm at subinhibitory concentrations.
Overall, I believe it is a well-written and well-organized article. However, I don't think the novelty provided is sufficient for publication in IJMS. In my opinion, antibiofilm studies would be necessary to complete the antimicrobial study of this selection of flavonoids.
Comments on the Quality of English LanguageAccording to the quality of the english language used, I think that in general terms it is good enough. However, it could be improved a bit along the text. There are some examples of sentences that do not sound quite right to me:
1. Sentence 19-20: ... scientifically validated methods of the antibacterial and antioxidant activities...
2. Sentence 77-81 is a bit large sentence
3. Sentence 132. Of the flavonoids... For me sound better: Among the flavonoids
Author Response
Response to Reviewer 2 Comments
- Summary
Thank you very much for taking the time to review this manuscript. Please find the detailed responses below and the corresponding corrections highlighted and in track changes in the re-submitted files.
All suggestions and comments were duly considered as follows:
R2: In my opinion, the synthetic part used in this article is widely known, and the novelty provided at this point is somewhat limited. On the other hand, the structural characterization of all the synthesized final compounds and intermediate chalcones is adequately described in the supplementary material, with high purity of all compounds confirmed by NMR and HRMS for the new compounds.
The antimicrobial activity of the synthesized compounds is where the main novelty of this article lies. The percentage of inhibition at 50 μg/mL along with their MIC against three bacteria, MRSA 97-7, E. coli, and L. monocytogenes, is described, with positive results found for compounds FV1, FV2, FV6, FO3, and FO10. Despite being promising results, in my opinion, it would be necessary to delve a little deeper into these antimicrobial activity results by conducting some antibiofilm studies, both inhibition and disruption of the preformed biofilm at subinhibitory concentrations.
Overall, I believe it is a well-written and well-organized article. However, I don't think the novelty provided is sufficient for publication in IJMS. In my opinion, antibiofilm studies would be necessary to complete the antimicrobial study of this selection of flavonoids.
Response: We believed that this work is considered sufficiently novel for the journal, based on the findings for both molecules with natural-like and novel structures. The introduction of a new SCH3 group was of interest from a biological point of view, and extending these modifications to other groups such as C-prenylated groups would enhance these activities. However, further studies are required to assess the antibiofilm and ROS activities of these compounds, which will be conducted in the medium term with other projects.
RV 2: 1. Sentence 19-20: ... scientifically validated methods of the antibacterial and antioxidant activities...
Response: The sentence has been re-written. Line 19-21
RV 2: 2. Sentence 77-81 is a bit large sentence
Response: The sentence has been re-written. Line 79-82
RV 2: 3. Sentence 132. Of the flavonoids... For me sound better: Among the flavonoids
Response: The sentence was changed. Line 124
Best Regards
Dr. Mauricio Osorio
Reviewer 3 Report
Comments and Suggestions for Authors
Review of the article "Antibacterial and antioxidant activity of synthetic polyoxygenated flavonoids" by Osorio-Olivares et al.
Well thought out article. The authors not only synthesize Ali compounds and study their antibacterial and antioxidant properties. It is a pity that the authors did not attempt to test the antioxidant properties using other methods (only the DPPH method was chosen) and did not conduct anticancer tests - especially for new compounds. Very well-prepared part of the spectroscopic characteristics included in SM.
Major remarks:
1) On what basis did the authors select certain compounds for synthesis? I miss the literary background. Please add this in the introduction.
2) Figure 1. It is very unclear to replace the C2-C3 bond with an X. I would suggest 2 separate structural formulas - one for flavanones and the other for flavones.
3) Line 65 - should be "and" instead of "y"
4) Line 74 - please quote correctly instead of "(Kim et al., 2022). There are also similar mistakes later in the article.
5) Line 84/85 - please write specifically which 4 compounds are new.
6) In compounds Fv1-FV11 - what was the configuration on carbon C2?
7) Why did the authors limit their antimicrobial tests to testing the lowest concentrations of 50 ug/mL? Particularly in cases where growth inhibition was low?
8) How do the authors explain MIC values for E. coli in literature citation [22] significantly higher for the FO10 compound compared to the data obtained as a result of their own research presented in the reviewed manuscript (Lines 171-172 vs Table 4)
9) Why didn't the authors test the antioxidant properties using other methods, e.g. FRAP, ABTS etc.? Possibly the selected mechanism for scavenging the DPPH free radical is inappropriate for this type of compounds?
10) Section 3.1.1. Please add manufacturers, cities and countries of the equipment used (where missing, of course).
11) Line 255 - "stand at room temperature" - why was the reaction not mixed?
Comments on the Quality of English LanguageMinor editing is required.
Author Response
Response to Reviewer 3 Comments
- Summary
Thank you very much for taking the time to review this manuscript. Please find the detailed responses below and the corresponding corrections highlighted and in track changes in the re-submitted files.
All suggestions and comments were duly considered as follows:
RV 3: 1) On what basis did the authors select certain compounds for synthesis? I miss the literary background. Please add this in the introduction.
Response: The selection of these molecules was based on previous research conducted by our team (ref. 17).
RV 3: 2) Figure 1. It is very unclear to replace the C2-C3 bond with an X. I would suggest 2 separate structural formulas - one for flavanones and the other for flavones.
Response: The drawing in fig.1 was modified as suggested by reviewers.
RV 3: 3) Line 65 - should be "and" instead of "y"
Response: Now modified in Line 67
RV 3: 4) Line 74 - please quote correctly instead of "(Kim et al., 2022). There are also similar mistakes later in the article.
Response: Now removed in line 75.
RV 3: 5) Line 84/85 - please write specifically which 4 compounds are new.
Response: Sorry, the new compounds are five FV6, FV7, FV9, FO6, and FV7
RV 3: 6) In compounds Fv1-FV11 - what was the configuration on carbon C2?
Response: The synthesis of the flavanones was carried out without stereochemical control, so the flavanones were obtained as racemic mixtures.
RV 3: 7) Why did the authors limit their antimicrobial tests to testing the lowest concentrations of 50 ug/mL? Particularly in cases where growth inhibition was low?
Response: This study employs a value of 50 µg/mL as a limit in order to facilitate comparisons with commercial antibiotics whose respective resistance values against these bacteria are close to 50 µg/mL as established by the Clinical and Laboratory Standards Institute (CLSI).
RV 3: 8) How do the authors explain MIC values for E. coli in literature citation [22] significantly higher for the FO10 compound compared to the data obtained as a result of their own research presented in the reviewed manuscript (Lines 171-172 vs Table 4)
Response: It is possible for different strains of the same bacterium to be affected in different ways by the same compound.
RV 3: 9) Why didn't the authors test the antioxidant properties using other methods, e.g. FRAP, ABTS etc.? Possibly the selected mechanism for scavenging the DPPH free radical is inappropriate for this type of compounds?
Response: A further objective is to evaluate other antioxidant methods with flavonoids that exhibit greater bioavailability, such as c-prenylated. However, these are part of other ongoing research initiatives.
RV 3: 10) Section 3.1.1. Please add manufacturers, cities and countries of the equipment used (where missing, of course).
Response: The missing data were duly incorporated.
RV 3: 11) Line 255 - "stand at room temperature" - why was the reaction not mixed?
Response: The sentence was changed by “The mixture was stirred at room temperature for a period of 24 hours”, line 252.
Best Regards
Dr. Mauricio Osorio
Round 2
Reviewer 2 Report
Comments and Suggestions for Authors
I would like to thank authors for accept my suggestions. It is true that perform the antiobiofilm studies could delay the publication of the article if the study is not ready in the laboratory. In any case, I think that carrying out these studies may be of interest. I'm glad to hear that author are considering doing them in the near future.
Reviewer 3 Report
Comments and Suggestions for Authors
Thank you for all the explanations and changes introduced.
If the Editor decides that Figure 1 can be presented in the form as presented, then in my opinion the manuscript meets all the criteria to be published in IJMS MDPI.
Comments on the Quality of English LanguageMinor editing is required at the final stage.